# Urinary MicroRNAs as Potential Markers for Non-Invasive Diagnosis of Bladder Cancer

**DOI:** 10.3390/ijms21113814

**Published:** 2020-05-27

**Authors:** Kati Erdmann, Karsten Salomo, Anna Klimova, Ulrike Heberling, Andrea Lohse-Fischer, Romy Fuehrer, Christian Thomas, Ingo Roeder, Michael Froehner, Manfred P. Wirth, Susanne Fuessel

**Affiliations:** 1Department of Urology, Technische Universität Dresden, 01307 Dresden, Germany; kati.erdmann@uniklinikum-dresden.de (K.E.); karsten.salomo@googlemail.com (K.S.); ulrike.heberling@uniklinikum-dresden.de (U.H.); andrea.lohse-fischer@uniklinikum-dresden.de (A.L.-F.); romy.fuehrer@uniklinikum-dresden.de (R.F.); christian.thomas@uniklinikum-dresden.de (C.T.); michael.froehner@ediacon.de (M.F.); manfred.wirth@uniklinikum-dresden.de (M.P.W.); 2National Center for Tumor Diseases (NCT), 01307 Dresden, Germany; anna.klimova@nct-dresden.de (A.K.); ingo.roeder@tu-dresden.de (I.R.); 3Institute for Medical Informatics and Biometrics, Technische Universität Dresden, 01307 Dresden, Germany

**Keywords:** miRNA, quantitative PCR, tumor marker, urothelial carcinoma, voided urine cytology

## Abstract

Currently, voided urine cytology (VUC) serves as the gold standard for the detection of bladder cancer (BCa) in urine. Despite its high specificity, VUC has shortcomings in terms of sensitivity. Therefore, alternative biomarkers are being searched, which might overcome these disadvantages as a useful adjunct to VUC. The aim of this study was to evaluate the diagnostic potential of the urinary levels of selected microRNAs (miRs), which might represent such alternative biomarkers due to their BCa-specific expression. Expression levels of nine BCa-associated microRNAs (miR-21, -96, -125b, -126, -145, -183, -205, -210, -221) were assessed by quantitative PCR in urine sediments from 104 patients with primary BCa and 46 control subjects. Receiver operating characteristic (ROC) curve analyses revealed a diagnostic potential for miR-96, -125b, -126, -145, -183, and -221 with area under the curve (AUC) values between 0.605 and 0.772. The combination of the four best candidates resulted in sensitivity, specificity, positive and negative predictive values (NPV), and accuracy of 73.1%, 95.7%, 97.4%, 61.1%, and 80.0%, respectively. Combined with VUC, sensitivity and NPV could be increased by nearly 8%, each surpassing the performance of VUC alone. The present findings suggested a diagnostic potential of miR-125b, -145, -183, and -221 in combination with VUC for non-invasive detection of BCa in urine.

## 1. Introduction

Bladder cancer (BCa) was the 10th most common malignancy worldwide with 550,000 new cases and an estimated 200,000 deaths in 2018 [1,2]. The highest BCa incidence rates in the world were reported for Southern and Western Europe and North America. In men, who are four times more frequently affected than women, BCa ranks sixth of all newly diagnosed cancers and is the ninth most deadly cancer worldwide [2]. Initial BCa diagnostic steps include voided urine cytology (VUC), as the current standard for non-invasive BCa detection, and cystoscopy, an invasive procedure. Tissue specimens obtained during transurethral resection of the bladder (TUR-B) undergo histopathological examination for assessment of the putative tumor grade and stage [3]. The same approach is used for regular follow-up after treatment to detect BCa recurrence and progression. Among other factors, these life-long, frequent, and cost-intensive surveillance procedures make BCa one of the most expensive tumor entities [4,5].

VUC is an observer-dependent method and characterized by a high specificity of 78–100%. While it displays a moderate sensitivity of 34–84% for high-grade tumors, its sensitivity for low-grade tumors is very low at 12–26% [6]. Even though several alternative biomarker-based tests for non-invasive BCa detection in urine exist, none of them is recommended for routine use in clinical practice due to their inadequate performance [3,7]. The sensitivity of most alternative BCa tests is higher compared to VUC, but their specificity is not able to reach that of VUC so far [8,9]. Therefore, the search for cost-efficient, highly specific, and sensitive biomarkers for non-invasive BCa diagnosis, screening, and follow-up, which eventually allow reducing the number of invasive, inconvenient, and expensive cystoscopies, is ongoing.

Numerous studies have been aimed at the identification of mRNA expression signatures in tumor tissues and tumor-derived urine specimens that reflect the presence and aggressiveness of BCa and additionally characterize the most powerful predictive biomarker combinations [10,11,12]. In recent years, microRNAs (miRNAs or miRs) emerged as potential diagnostic and prognostic biomarkers due to their tissue- and tumor-specific expression. miRNAs are small, endogenous, non-coding RNAs that function as post-transcriptional regulators of specific target genes via mRNA degradation or inhibition of the translation [13,14]. As regulators of many cellular processes such as differentiation, proliferation, apoptosis or cell cycle control, miRNAs also play an important role in different deregulated, pathological pathways such as tumor onset and progression, making them promising candidates as tumor markers [13,14,15].

BCa-associated miRNA signatures were investigated by microarray, next-generation sequencing (NGS), and quantitative PCR (qPCR) analyses in malignant and non-malignant bladder tissue specimens, as well as in urine samples from BCa patients and control subjects. Several promising miRNA candidates showing a differential expression and/or an association with BCa aggressiveness emerged from these studies [16,17,18,19,20]. Additionally, the proven involvement of many of these miRNAs in tumor-related pathways deems them worthy of evaluation as potential biomarkers for diagnosis of BCa [21]. On the basis of a literature search comprising single reports, systematic reviews, and meta-analyses, nine miRNAs fulfilling the abovementioned criteria were selected for an independent validation study as putative biomarkers for non-invasive BCa detection in urine specimens: miR-21, -96, -125b, -126, -145, -183, -205, -210, and -221 [22,23,24,25,26,27,28,29,30,31,32] (for details, see also Appendix A). Using urine sediments from BCa patients and control subjects, the expression levels of these miRNAs determined by qPCR were assessed individually and in combination with regard to their diagnostic potential. In order to determine a potential diagnostic improvement, their diagnostic performance was also evaluated as an adjunct to VUC.

## 2. Results

### 2.1. Characteristics of BCa Patients and Control Subjects

A total of 104 patients with histologically-proven BCa at TUR-B were included in the present study (Table 1). This cohort consisted of 83 male and 21 female patients with a median age of 70 years (range 50 to 85 years). The relative distribution of pTa, pT1, pTis, and ≥pT2 tumors was 48.1%, 21.2%, 14.4%, and 16.3%. Since all of the 15 pTis tumors occurred concomitantly with other tumors, the most severe tumor stage was coded. In doing so, patients with non-muscle-invasive BCa (NMIBC; pTa and pT1) and concomitant pTis were allocated to the pTis group (10 of the 15 pTis cases; 9.6% of all BCa). The remaining five patients with muscle-invasive BCa (MIBC) and concomitant pTis were allocated to the group ≥pT2, resulting in a total of 22 patients with MIBC with or without concomitant pTis (21.1% of all BCa). Multifocal tumors occurred in 32 patients (30.8%).

Using the WHO grading system from 1973, 13.5% G1, 50.0% G2, and 36.5% G3 BCa were reported. According to the WHO grading system from 2004, 83.7% of the BCa patients displayed high-grade tumors, whereas only 16.3% had low-grade tumors.

Moreover, the control group consisted of eight subjects, who underwent a TUR-B without histopathological evidence of BCa and 38 patients with urolithiasis (Table 2).

### 2.2. Correlations between the Analyzed miRNAs

Relative expression levels of the selected miRNAs (determined by qPCR and normalized to the geometric mean of the reference RNAs RNU44 and RNU48) were analyzed in detail in BCa-derived and control urine sediments. Some of the analyzed miRNAs showed significant correlations with each other as assessed by the pairwise calculation of Spearman’s rank correlation coefficients. Most prominent significant positive correlations (*p* < 0.01) were observed between the miRNAs miR-21, -125b, -205, -210, and -221 with correlation coefficients between 0.566 (miR-210 vs. miR-221) and 0.825 (miR-125b vs. miR-221). Significant positive correlations (*p* < 0.01) were also seen between miR-96 and miR-183 (*r_s_* = 0.557), miR-96 and miR-126 (*r_s_* = 0.311), and miR-126 and miR-183 (*r_s_* = 0.483). Finally, miR-145 correlated negatively (*p* < 0.01) with miR-205 (*r_s_* = −0.433) and miR-210 (*r_s_* = −0.483).

### 2.3. Associations of Urinary miRNA Transcript Levels and Histopathological Features

The expression levels of miR-125b (*r_s_* = −0.434, *p* < 0.01) and miR-205 (*r_s_* = −0.206, *p* < 0.05) in the tumor-derived urine samples decreased with increasing tumor stage, whereas those of miR-96 and miR-183 showed a weak positive correlation with tumor stage (*r_s_* = 0.238 and 0.246, *p* < 0.05). A weak, but significant negative correlation (*p* < 0.01) with tumor grade according to the WHO classification from 1973 was detected for miR-125b, miR-205, and miR-210 (*r_s_* between −0.309 and −0.334), whereas miR-145 displayed a weak, but significant positive correlation with tumor grade (*r_s_* = 0.266, *p* < 0.01). A similar trend of dependence on tumor grade assessed according to the WHO classification from 2004 could only be observed for miR-125b and -210. The inverse relationships between the miR-125b expression levels in urine and tumor stage, as well as with tumor grade are exemplarily shown in Figure 1. Significant differences in relative miR-125b levels were found between the controls and all tumor stages, as well as between pTa tumors and higher tumor stages. Moreover, relative miR-125b levels differed significantly between the controls and G2/G3 or high-grade BCa, as well as between G1 and G2/G3 or low-grade and high-grade tumors (Figure 1). None of the miRNAs displayed substantial alterations between urine samples from patients with unifocal and multifocal BCa.

### 2.4. Evaluation of the Single miRNAs as Potential Markers for BCa Detection

Next, the relative miRNA expression levels were compared between the urine samples from the eight control patients without histopathological evidence of BCa at TUR-B and from the 38 controls patients with urolithiasis. None of the nine analyzed miRNAs showed significant differences in relative expression levels between both control subgroups (Mann–Whitney U test: *p* > 0.05). Therefore, data from both subgroups were combined, constituting the final control group of 46 subjects.

In comparison to these controls, statistically significant different relative miRNA expression levels were observed in urine sediments from BCa patients for miR-96, -125b, -126, -145, -183, and -221. After Bonferroni’s correction for multiple comparisons (Table 3 and Figure 2), the significance was retained for these miRNAs except for miR-96 (*p* = 0.05). While the miRNAs miR-96, -126, and -183 displayed higher relative expression levels in the BCa-derived urine sediments, the miRNAs miR-125b, -145, and -221 were downregulated in these specimens compared to the control group (Table 3). Receiver operating characteristic (ROC) curve analyses also revealed a diagnostic potential for these six miRNAs with area under the curve (AUC) values ranging from 0.605 (*p* = 0.369) for miR-96 to 0.772 (*p* < 0.001) for miR-221 (Table 3 and Figure 3). The remaining three miRNAs miR-21, -205, and -210 did not show significant differences between tumor and control subjects (Table 3).

### 2.5. Comparison of Diagnostic Performance of miRNA Transcript Levels and Voided Urine Cytology

To assess the diagnostic power of the single miRNAs, VUC, and combinations thereof, different statistical approaches were applied. The first approach was based on the simple classification of the measured relative miRNA expression levels and of the results of VUC assessment into positive (“1”) and negative (“0”) indicators of tumor detection. The diagnostic parameters calculated on this basis were assessed with regard to the best performing single BCa marker. The nine analyzed miRNAs displayed only moderate diagnostic value with accuracy rates between 48.7% for miR-96 and 78.0% for miR-125b (Table 4). None of these single miRNAs could surpass the performance of VUC, which displayed a relatively high sensitivity of 76.9% at a specificity of 100% and an accuracy of 84.0% (Table 4) in our cohort.

In the next step, the tumor detection values (“1” or “0”) for the six miRNAs with substantially different relative expression levels between urine samples from BCa patients and controls were totaled and evaluated with regard to the best separation of these sums. In doing so, the detection of at least four of these six miRNAs as an indicator of the tumor yielded a sensitivity of 73.1%, a specificity of 93.5%, and an accuracy of 79.3% (Table 5). The combination of only the four miRNAs with the highest AUC values in ROC curve analyses (miR-125b, -145, -183, and -221) and the use of at least three positive miRNAs as a BCa indicator resulted in a slightly better diagnostic performance compared to the combination of six miRNAs and to all single miRNAs (Table 4 and Table 5).

The inclusion of VUC in both miRNA combinations led to a further increase of the diagnostic power and revealed the combination of the four miRNAs with VUC as the best approach with a sensitivity of 84.6%, a specificity of 95.7%, and an accuracy of 88.0% (Table 5). This combination was able to outperform VUC as a single marker with a 4% increase in accuracy, but at the expense of specificity in the same magnitude. However, it also resulted in a gain of nearly 8% each in sensitivity and NPV.

In the second statistical approach, the diagnostic power was analyzed using penalized linear regression applied to standardized miRNA values. This regression analysis (Figure 4) indicated that miR-210, -125b, and -221 had the strongest diagnostic power, while the influence of miR-145 and -205 was much weaker and the one of miR-21, -126, and -183 the weakest. In 10,000 simulations, each performed for a sample set of 20 patients from the original dataset, the penalized regression model correctly classified the tumor status in about 75% of the cases.

The order of importance of the miRNAs (miR-125b, -221, -145, -183, -205, -126, -210, -21) obtained using the random forest method as the third statistical approach was in concordance with the results of penalized linear regression. The percentage of cases correctly classified with this approach was 82.6%. The classification error in the random forest analysis was 20% (based on 100 simulations). Because of too many observed zero values for miR-96 in the control group, this variable was excluded from the analyses with *glmnet* and *randomForest*. In the overview of all statistical evaluation methods used herein, the miRNAs miR-125b, -145, -183, and -221 appeared to serve as well-suited biomarkers for the non-invasive detection of BCa in urine sediments independent of the chosen approach.

## 3. Discussion

The early and reliable detection of BCa is one of the biggest challenges to identify and treat patients at risk and to commence treatment in a timely manner. Many efforts have been made in recent years to implement new diagnostic approaches in clinical routine for the non-invasive detection of BCa using urine [6,8,9] and to replace VUC as the current gold standard. Nevertheless, none of these new biomarkers is thus far recommended for routine use in the early detection or follow-up of BCa or for replacement or reduction of the invasive procedure of cystoscopy [3]. According to current guidelines, such urinary molecular marker tests should provide a very high NPV to predict the absence of tumor reliably and eventually to avoid unnecessary cystoscopies [3]. Moreover, they should amend and strengthen the diagnostic performance of VUC and additionally overcome its main drawback: the low sensitivity, particularly for low-grade BCa [6].

For this purpose, this study focused on the diagnostic capability of miRNAs which are well known to be dysregulated in cellular pathways associated with the onset and progression of tumors including BCa [16,18,20]. Based on an extensive literature search, a number of promising miRNA candidates was selected, which were reported to be differentially expressed in bladder tissues and/or urine specimens from BCa patients and control subjects [22,23,24,25,26,27,28,29,30,31,32]. The diagnostic performance of the altered expression of these miRNAs in urine sediments was evaluated, revealing a certain potential of the miRNAs miR-96, -125b, -126, -145, -183, and -221 as single markers or in combination. The best diagnostic power was finally achieved by the use of four of these miRNAs in combination with VUC by simple classification of the urine samples as positive or negative for the single markers and subsequent allocation according to the total number of altered single markers (0-2 vs. 3-5 positive markers) regardless of which markers were changed. This approach resulted in a sensitivity of 84.6%, an NPV of 73.3%, and an accuracy of 88.0%, all surpassing the corresponding parameters of VUC alone (76.9%/65.7%/84.0%). In the marker combination, this was accompanied by losses of 4% in specificity and 2% in PPV compared to VUC alone where both accounted for 100%. Nevertheless, these losses appeared less severe in the face of the gain of nearly 8% each in sensitivity and NPV.

Thus, the combination of the four miRNAs miR-125b, -145, -183, and -221 with VUC seemed to have a promising potential for reliable diagnosis of BCa in urine samples prior to TUR-B. Accordingly, the diagnostic value of a reduced miR-125b expression in urine to prior tumor resection previously reported by Snowdon et al. [33], Mengual et al. [29], Zhang et al. [32], and Pospisilova et al. [34] could be confirmed in this study. This miRNA displayed a steady decrease with increasing tumor stage and grade, as also observed by Zhang et al. [32], indicating a certain prognostic power. Accordingly, Mengual et al. [29] described a prognostic panel comprising miR-125b and -92a for the identification of high-grade BCa, which was, however, not assessed in our patient cohort.

Another miRNA frequently described to be downregulated in cancer and particularly in BCa is miR-145 [27,28,35,36], which was also one of the best single miRNA markers in this study. Its diagnostic power (AUC = 0.687) herein was comparable to that (AUC = 0.729) determined by Yun et al. [36]. In contrast, Mearini et al. did not observe significant differences in urinary miR-145 levels between urine sediments from BCa patients and control subjects [37].

The miRNAs miR-96 and miR-183 are encoded by the miR-183/96/182 cluster on chromosome 7 and regulated in a concerted manner in different pathologies including BCa and other cancers [27,38,39,40]. Therefore, their expression behavior appears to be very similar as also reflected by their correlation observed in our measurements. Both miRNAs displayed a substantial elevation in BCa-derived urine samples compared to controls with miR-183 showing the second-best diagnostic potential of all miRNAs investigated herein (AUC = 0.720). Yamada et al. analyzed urinary levels of miR-96 and miR-183 in patients with urothelial carcinoma and revealed a high diagnostic value for both miRNAs reflected by AUC values of 0.831 and 0.817, respectively [41]. Moreover, they observed a positive dependence of miR-96/-183 levels on tumor stage and grade, whereas we could only identify a weak, but significant positive correlation with tumor stage. Considerable diagnostic power with regard to the non-invasive diagnosis of BCa by determination of urinary miR-96 levels was also described in several other studies [42,43,44]. This miRNA showed a differential expression only with a statistical trend (*p* = 0.05) in our study, but had to be excluded from some statistical approaches due to its low expression levels. miR-183, which was better performing in our cohort than miR-96, also emerged as a possible marker for high-grade NMIBC in the report from Pardini et al., who identified promising diagnostic miRNAs by NGS in cell-free urine supernatants from BCa patients and control subjects [45]. In summary, from these data, miR-96 and/or miR-183 seemed to provide substantial information when analyzing urine specimens for BCa diagnosis.

The diagnostic usability of elevated levels of miR-126 as a urinary marker has been reported by several studies so far [33,44,46]. In 2010, Hanke et al. reported an AUC value of 0.768 for the separation of BCa patients vs. healthy donors and of 0.747 vs. healthy donors and infections for miR-126 detection in urine [46]. Interestingly, miR-126 and -183 showed a significant, but less strong positive correlation with each other than miR-96 and -183, but they were both useful as BCa markers in contrast to miR-96.

Remarkably, miR-221 (our best performing BCa marker (AUC = 0.772)) was not found to be a promising candidate in previous studies despite its reported differential expression in BCa tissues [44,47]. In contrast, the remaining candidates (miR-21, -205, and -210), which did not show differential urinary expression levels in this study, were described as promising urinary BCa markers in selected reports [43,45,48,49,50]. For example, Michaildi et al. obtained a diagnostic AUC value of 0.845 when comparing miR-205 levels in cell-free urine supernatants from 177 normal donors and 32 BCa patients [50]. This miRNA also seemed promising with regard to the distinction of different BCa subsets as reported by Pardini et al. [45]. In contrast, it did not allow discrimination between patients with and without BCa, neither in the urine sediment, nor in the cell-free urine supernatant, in the study performed by Wang et al. [51]. Kim et al., who selected eight miRNAs (including miR-96, -125b, -145, and -205, like in our study) for urine analyses, observed among others increased levels of miR-205, as well as decreased levels of miR-125b and -145 in BCa-derived urine samples compared to negative controls [52]. We observed strong positive correlations between the miRNAs miR-21, -125b, -205, -210, and -221, but only miR-125b and -221 proved to be useful BCa markers in our study, for which we have no conclusive explanation. At the functional level, all of the five mentioned miRNAs were reported to regulate genes involved in cell cycle control, p53, and growth factor receptor signaling, which would fit with the observed correlations [53,54].

To explain the discrepancies observed between different studies, a variety of reasons can be given. On the one hand, there is a large diversity in the type of urinary components explored for miRNA analysis ranging from whole urine over urine sediment as used herein to cell-free urine supernatant or exosomes (Appendix A). It was assumed that exfoliated tumor cells were the main constituents of cellular urine sediments from BCa patients, and miRNA levels measured therein directly reflected those of the tumor. Both increased and decreased expression of such miRNAs would indicate the presence of BCa against the background of miRNA patterns in non-malignant cells, which are also part of the urine sediment. On the other hand, the kind of miRNA determination varied from NGS- and microarray-based profiling to different kinds of qPCR with and without miRNA-specific probes and the possible inclusion of a preamplification step. Moreover, the number of included patients and controls, the kind of controls (healthy donors or patients with benign urological diseases, infection, or hematuria), and tumor subtypes (e.g., urothelial carcinoma of the bladder or the upper urinary tract, bilharziasis-associated BCa) profoundly influenced the results obtained.

As revealed in this and many other studies, marker combinations seem to be superior to the analysis of single markers. The diagnostic gain, particularly in comparison to VUC alone, was clearly shown herein by the combination of VUC with the best four miRNAs as discussed above. In analogy, Yamada et al. and Eissa et al. reached an increase in diagnostic power by the combined analysis of VUC and miR-96 due to a considerable number of non-overlapping cancer patients who were positive only for one of these markers. This was particularly reflected by an increase in sensitivity, NPV, and overall accuracy [41,42]. The same was true for the combination of VUC and more miRNAs, as shown by Eissa et al. for miR-10b/-29c/-210 [47], as well as in our study for miR-125b/-145/-183/-221.

However, it remains to be verified whether the achieved improvement of the diagnostic performance of VUC by combination with miRNA levels in urine sediment would really be sufficient to reduce cystoscopies for the first diagnosis or for surveillance after therapy. Alternatively, other BCa-specific biomarkers could potentially fulfill these requirements. Meanwhile, several promising tests became commercially available that detect altered biomarkers in exfoliated tumor cells such as *Xpert Bladder Cancer Monitor*, *Bladder EpiCheck test*, or *AssureMDx test* (reviewed in [55]). The *Xpert test*, which is based on differential mRNA patterns in BCa cells, delivered a sensitivity of 74% and a specificity of 80% for surveillance of NMIBC patients [56]. This was comparable to the results of the *EpiCheck test*, which reflects altered DNA methylation in BCa cells, with a sensitivity of 68% and a specificity of 88% for the detection of recurrent NMIBC [57]. Similar results were reported by Trenti et al., who directly compared the two tests with VUC in the follow-up of NMIBC patients [58]. The *AssureMDx test* (a combined DNA methylation and mutation urine assay) was reported to serve as a useful diagnostic tool to select patients with hematuria for cystoscopy with a sensitivity of 93% and a specificity of 86%, indicating the highest diagnostic potential among the new commercially available biomarker tests [59,60]. Thus, these tests seemed to provide a similar diagnostic power as our test, but only a head-to-head comparison in defined patient cohorts would deliver conclusive results with regard to first BCa diagnosis or surveillance.

Another factor that decisively determines the significance and predictive power of calculated marker combinations is the kind of statistical analysis methods. First, we compared the simple classification of urine samples as positive or negative for the single markers and subsequent allocation according to the total number of altered single markers regardless of which markers were changed with a penalized linear regression. Interestingly, penalized regression indicated the strongest diagnostic power for miR-210, -125b, and -221, although miR-210 was not identified as a suitable marker by ROC analyses. Nevertheless, the diagnostic accuracy of 79.3% and 80.0% for the combination of six or four miRNAs, respectively, accomplished by the first calculation method was higher than that of 75% yielded by penalized linear regression. The random forest method revealed a comparable accuracy of 82.6% using the same miRNAs as our first classification method. Additional principle component analyses could not further increase the diagnostic performance of different miRNA combinations in our study, but should be kept in mind as a further alternative approach, as well as the decision-tree analyses reported by Snowdon et al. and Pospisilova et al. [33,34].

Finally, the general approach for the selection of miRNAs as potential diagnostic BCa markers seems worthy of discussion, as well. The most comprehensive, but also the most cost- and labor-intensive approach is whole transcriptome profiling by NGS, microarray, or PCR array analyses followed by validation of the best candidates by independent methods such as qPCR in independent cohorts [29,32,34,44,45,48,49]. Several other authors made the choice in the same way as in the present study on the basis of publicly accessible datasets and literature reports on differential miRNA expression in malignant and non-malignant bladder tissues and/or in urine samples from BCa patients and suitable controls [33,36,51]. This approach primarily serves the independent validation of candidates that have already been investigated and shown to be promising, supporting the identification of the best markers. On the basis of our results and the presented literature review (Appendix A), the miRNAs miR-125b and -183 could represent such promising BCa-associated markers in urine. However, further miRNA candidates with a reported considerable diagnostic potential comprised of miR-99a [32,34,52], miR-146a [61], miR-155 [62], or members of the miR-200 family [30,36,51,52] should be further evaluated in comparative analyses.

Our study was associated with several limitations including the relatively low number of control subjects, since such patients are much less common in our department than BCa patients. However, the implementation of patients with benign urological diseases as done herein seemed to be more suitable than the comparison to healthy controls without any symptoms and with fewer exfoliated cells in urine sediment. Furthermore, the BCa patient cohort analyzed in the present study was comprised of a high percentage of cases with high-grade tumors, which is typical for specialized urological clinics and not comparable to those of private urological practices.

Considering this drawback, we avoided extensive calculations of possible associations with tumor grade. However, studies including more low-grade BCa would be required to assess the usefulness of this approach to identify these patients reliably. The clear advantages of our study were the direct comparison to VUC as the current gold standard for non-invasive BCa detection in urine and the combination of miRNA markers with VUC, which resulted in a superior diagnostic power. Nevertheless, this has to be confirmed in future studies on more BCa patients at different stages and suitable controls.

## 4. Materials and Methods

### 4.1. Study Population, Data, and Sample Collection

The present study was approved by the institutional review board of the Medical Faculty at the Technische Universität Dresden. Written informed consent was obtained from every participant. Patients with suspicion of having BCa and control subjects were prospectively recruited between May 2014 and September 2016. The inclusion criteria for the cases were comprised of new-onset BCa diagnosed at TUR-B and an age between 40 and 85 years. Patients diagnosed with papilloma or papillary urothelial neoplasm of low malignant potential (PUNLMP) were not eligible. In total, one-hundred four patients undergoing TUR-B with histologically-proven BCa were included (Table 1).

Eight patients suspected to have BCa, but histopathologically diagnosed as tumor-free, were allocated to the control group. Additionally, thirty-eight patients with urolithiasis were included as controls, finally resulting in a total control group size of 46 subjects (Table 2).

The histopathological examination of the resected bladder specimens, which served as the reference standard for BCa diagnosis, was performed using the UICC TNM classification from 2011 [63]. The tumors were accordingly classified as non-muscle-invasive BCa (NMIBC), comprised of the tumor stages pTa, pT1, and pTis, or as muscle-invasive BCa (MIBC), including all tumors with an expected tumor stage of ≥pT2a. The stratification of tumor grades into G1, G2, and G3 was done in accordance with the WHO classification from 1973 and into low-grade and high-grade tumors according to the WHO classification from 2004 [64]. As a reference method for the non-invasive detection of BCa, VUC specimens were prepared from urine samples from all BCa patients and control subjects. One experienced examiner (U.H.) evaluated these VUC specimens according to the ICUD/WHO classification [65].

### 4.2. Processing of Urine Samples, RNA Isolation, and cDNA Synthesis

Spontaneous urine samples (20–100 mL) were collected before therapeutic intervention from all BCa patients and control subjects. Urine collection was standardized in daily clinical practice (no morning urine, 2nd or 3rd urine of the day, clean catch in a sterile cup). The presence of erythrocytes, leucocytes, and bacteria was assessed by microscopic analysis of the sediment from 5–10 mL urine. Additionally, a VUC specimen was prepared from 5–10 mL urine. It was prefixed with *Esposti’s fixative* overnight, centrifuged on glass slides, fixed with *Cytofix N* (Niepötter Labortechnik, Bürstadt, Germany), and stained by the Papanicolaou procedure [66].

The remaining urine was centrifuged at 1500× *g* for 10 min at 4 °C. After removal of the supernatant, the cellular pellet was washed twice with ice-cold phosphate-buffered saline (PBS) by centrifugation at 870× *g* for 5 min at 4 °C and resuspended in 700 µL *QIAzol Lysis Reagent* (Qiagen, Hilden, Germany). The lysates were frozen and stored at −80 °C until further processing. After thawing, total RNA was isolated using the *Direct-zol RNA MiniPrep kit* (Zymo Research, Freiburg, Germany) according to the manufacturer’s recommendations. The RNA was eluted from the *Zymo-Spin IIC Column* with 40 μL nuclease-free water and quantified using the *NanoDrop 2000c* spectrophotometer (PEQLAB, Erlangen, Germany) and the *Agilent RNA 6000 Pico Kit* on an *Agilent 2100 Bioanalyzer* (Agilent Technologies, Ratingen, Germany).

If available, one-hundred nanograms total RNA (at least 50 ng) were employed for reverse transcription (RT) of the miRNAs using specific *TaqMan microRNA Assays* (Thermo Fisher Scientific, Darmstadt, Germany).

A multiplex RT was performed for the nine selected target miRNAs (Table 6: miR-21, -96, -125b, -126, -145, -183, -205, -210, and -221) and two reference RNAs (RNU44 andRNU48) in a final reaction volume of 30 µL comprised of 8 µL of the diluted RNA and 22 µL of the RT master mix. The latter consisted of dNTPs (each in a final concentration of 2 mM), *MultiScribe Reverse Transcriptase* (300 U), RT buffer, RNase inhibitor (7.6 U), and the respective RT primers (each 0.6×). The following temperature program was applied for the RT reaction: 30 min at 16 °C, 30 min at 42 °C, 5 min at 85 °C, followed by cooling down to 4 °C.

### 4.3. Transcript Quantitation by Quantitative PCR

The individual mature miRNAs and reference RNAs were quantified separately on the *LightCycler 480 Real-Time PCR System* (Roche Diagnostics, Mannheim, Germany). Each qPCR with a final volume of 10 µL consisted of 1 µL of the undiluted cDNA product, the respective *TaqMan MicroRNA Assay* (Table 6), *GoTaq Probe qPCR Master Mix* (Promega, Mannheim, Germany), and nuclease-free water. The qPCR temperature program comprised the following steps: 10 min initial denaturation at 95 °C, 45 cycles of 15 s denaturation at 95 °C, and 1 min annealing/extension at 60 °C. The transcript quantitation was carried out in two independent reactions, and threshold cycles (C_T_) determined by the second derivative method were averaged for each transcript per sample. In cases of a mean deviation >0.25 C_T_-value, the measurements were repeated. Subsequently, the delta-delta-C_T_ method was used for the calculation of the relative miRNA levels normalized to the reference RNAs. For this, the geometric means of the C_T_-values of RNU44 and RNU48 were utilized.

### 4.4. Statistical Analysis

For the assessment of potential differences in the relative miRNA levels between BCa patients and control subjects, the nonparametric two-tailed Mann–Whitney U test was applied. Bonferroni’s correction for multiple comparisons was performed by multiplying the uncorrected *p*-values by a factor of 9 for nine miRNAs. Spearman’s rank correlation coefficients (r_s_) were calculated to reveal possible correlations among the relative expression levels of the different miRNAs and with tumor stage or grade. The distribution of the relative miRNA expression levels is depicted in box plots, where the bottom and top of the boxes represent the first and third quartiles, respectively. The median is shown as a solid line within the box, and the ends of the whiskers are depicted according to the Tukey method. Data outside the whiskers represent outliers and are marked as single circle symbols.

Differences in the relative miRNA levels between patients with different tumor stages and grades were assessed by the nonparametric two-tailed Mann–Whitney U test.

The diagnostic performance of the individual miRNAs was assessed by ROC curve analyses, followed by the above-mentioned Bonferroni’s correction for multiple comparisons and the corresponding AUC values. The Youden index was calculated to determine optimal cutoff values, which were used to classify the relative expression levels of the different miRNAs into positive and negative indicators for tumor detection. Subsequently, the sensitivity (SNS), specificity (SPC), positive predictive value (PPV), negative predictive value (NPV), positive likelihood ratio (pLR), negative likelihood ratio (nLR), and accuracy (ACC) were calculated for each miRNA marker and for marker combinations according to standard statistical methods. The same was done for VUC alone or in combination with different miRNAs. For all analyses, two-sided *p*-values < 0.05 were considered statistically significant. All analyses were performed using *IBM SPSS Statistics Version 24.0.0.2* (IBM, Ehningen, Germany) and *GraphPad Prism Version 6.05 for Windows* (GraphPad Software, Inc, San Diego, CA, USA).

In addition, the diagnostic performance was assessed using penalized linear regression with miRNAs as predictor variables and tumor status as the outcome. The analysis was performed using the *R statistical environment* [67]. The predictive strength of the miRNAs for tumor status was investigated using the elastic net method [68] implemented in the R package *glmnet*. The importance of the miRNAs in predicting the tumor status and prediction accuracy were also assessed using the random forest method [69] implemented in the R package *randomForest* [70].

## 5. Conclusions

The literature-based selection of promising miRNAs as potential markers for the non-invasive detection of BCa in urine as alternative or adjunct markers to VUC revealed four suitable candidates (miR-125b/-145/-183/-221). In combination with VUC, an adequate performance was obtained, and the previously reported value as BCa-associated biomarkers could be confirmed for these four miRNAs. Prospective studies are required in order to reveal the real value of miRNAs in urine-based BCa diagnosis as a potential tool for the reduction of invasive and expensive diagnostic procedures like cystoscopies.

## Figures and Tables

**Figure 1 ijms-21-03814-f001:**
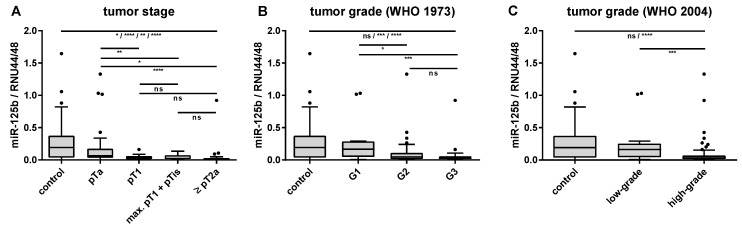
Dependence of the relative miR-125b expression levels (normalized to the geometric mean of the reference RNAs RNU44 and RNU48) in urine sediments on histopathological factors. The relationships between urinary miR-125b expression levels and tumor stage (**A**), as well as tumor grade according to the WHO classifications from 1973 (**B**) and from 2004 (**C**) are shown. Differences were tested by the Mann–Whitney U test. ns, not significant; * *p* < 0.05; ** *p* < 0.01; *** *p* < 0.001; **** *p* < 0.0001.

**Figure 2 ijms-21-03814-f002:**
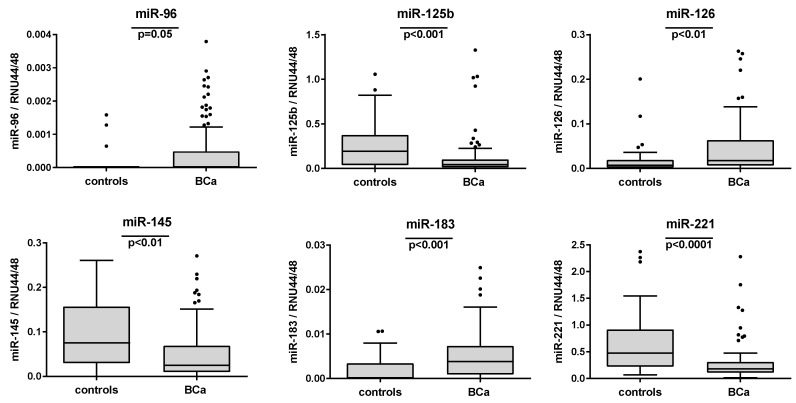
The distribution of the relative expression levels of the significantly altered miRNAs in urine sediments from controls and BCa patients is presented by box plots. Differences in the relative expression levels of the miRNAs (normalized to the geometric mean of the reference RNAs RNU44 and RNU48) were assessed using the Mann–Whitney U test followed by Bonferroni’s correction for multiple comparisons.

**Figure 3 ijms-21-03814-f003:**
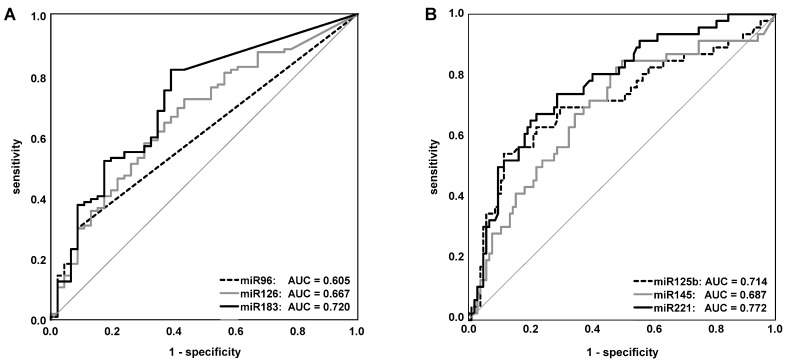
The diagnostic value assessed by ROC curve analysis of the six most promising miRNAs, which were significantly up- (**A**) or down-regulated (**B**) in urine sediments from BCa patients in comparison with controls. AUC, area under the curve; ROC, receiver operating characteristic.

**Figure 4 ijms-21-03814-f004:**
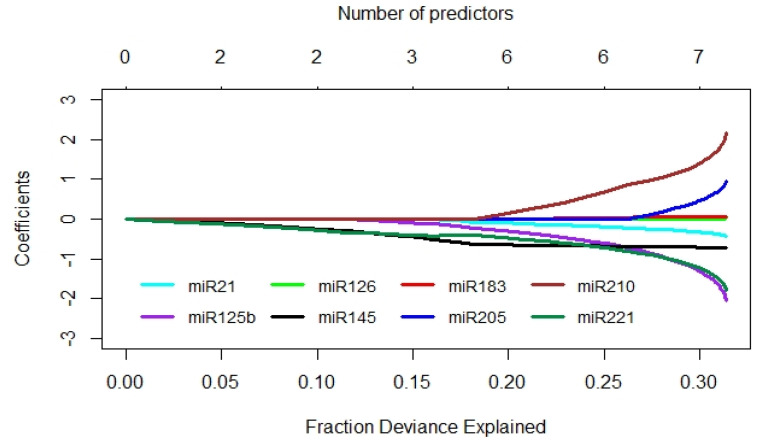
Diagnostic value of the miRNAs assessed by penalized linear regression. The deviance profile plot corresponding to standardized miRNAs as predictor variables in the *glmnet* model is shown. The slope of each path changes each time as another miRNA enters the model. The miRNAs, which explain a larger fraction of deviance, play a more important role in the prediction of the tumor status.

**Table 1 ijms-21-03814-t001:** Demographic, clinical, and histopathological characteristics of the BCa patients (*n* = 104). The table shows the absolute and relative distribution of gender, age, and clinicopathological parameters.

Parameter	Category	Number (*n*)	Percentage (%)
gender	male	83	79.8
	female	21	20.2
age ^1^ (years)	≤70.0	54	51.9
	>70.0	50	48.1
tumor stage	pTa	50	48.1
	pT1	22	21.2
	pTis	15	14.4
	*pTis only*	*0*	*0.0*
	*pTis + pTa*	*3*	*2.9*
	*pTis + pT1*	*7*	*6.7*
	*pTis + ≥ pT2a*	*5*	*4.8*
	≥ pT2a	17	16.3
tumor grade (WHO 1973)	G1	14	13.5
	G2	52	50.0
	G3	38	36.5
tumor grade (WHO 2004) multifocality	low-grade	17	16.3
	high-grade	87	83.7
	unifocal	72	69.2
	multifocal	32	30.8
voided urine	positive	80	76.9
cytology	negative	24	23.1

^1^ Age was dichotomized at the median (70.0 years).

**Table 2 ijms-21-03814-t002:** Demographic, clinical, and histopathological characteristics of the control subjects. The control group was comprised of 46 subjects in total, whereupon eight patients were histopathologically negative for BCa at TUR-B and 38 patients had urolithiasis. The table shows the absolute and relative distribution of gender, age and clinicopathological parameters.

Parameter	Category	Number (*n*)	Percentage (%)
gender	male	30	65.2
	female	16	34.8
age ^1^ (years)	<64.5	23	50.0
	≥64.5	23	50.0
diagnosis	BCa-negative TUR-B	8	17.4
	urolithiasis	38	82.6
voided urine	positive	0	0.0
cytology	negative	46	100.0

^1^ Age was dichotomized at the median (64.5 years).

**Table 3 ijms-21-03814-t003:** Comparison of the relative miRNA expression levels in urine sediments from the BCa and control groups (Mann–Whitney U test) and assessment of the diagnostic power by ROC curve analyses. All *p*-values were adjusted by Bonferroni’s correction for multiple comparisons.

miRNA	Regulation	Mann–Whitney	ROC Curve Analysis	
	in BCa	U Test (*p*-Value)	AUC	*p*-Value
miR-21	not different	=1.000	0.581	=1.000
miR-96	up	=0.050	0.605	=0.369
miR-125b	down	<0.001	0.714	<0.001
miR-126	up	<0.01	0.667	<0.01
miR-145	down	<0.01	0.687	<0.01
miR-183	up	<0.001	0.720	<0.001
miR-205	not different	=1.000	0.537	=1.000
miR-210	not different	=1.000	0.526	=1.000
miR-221	down	<0.0001	0.772	<0.0001

95% CI, 95% confidence interval; AUC, area under the curve; ROC, receiver operating characteristic.

**Table 4 ijms-21-03814-t004:** Diagnostic performance of the miRNAs and VUC as single markers.

Parameter	miR-21	-96	-125b	-126	-145	-183	-205	-210	-221	VUC
SNS	0.865	0.298	0.885	0.885	0.500	0.817	0.779	0.663	0.779	0.769
SPC	0.304	0.913	0.543	0.217	0.848	0.609	0.435	0.500	0.674	1.000
PPV	0.738	0.886	0.814	0.719	0.881	0.825	0.757	0.750	0.844	1.000
NPV	0.500	0.365	0.676	0.455	0.429	0.596	0.465	0.397	0.574	0.657
pLR	1.244	3.428	1.938	1.130	3.286	2.089	1.378	1.327	2.388	n.d.
nLR	0.442	0.769	0.212	0.531	0.590	0.300	0.509	0.673	0.328	0.231
ACC	0.693	0.487	0.780	0.680	0.607	0.753	0.673	0.613	0.747	0.840

ACC, accuracy; n.d., not determinable (division by zero); nLR, negative likelihood ratio; NPV, negative predictive value; pLR, positive likelihood ratio; PPV, positive predictive value; SNS, sensitivity; SPC, specificity.

**Table 5 ijms-21-03814-t005:** Diagnostic performance of combinations of selected miRNAs with each other and with VUC.

Parameter	6 miRs 96/125b/126/145/183/221 0-3/4-6 pos. Markers	4 miRs 125b/145/183/221 0-2/3-4 pos. Markers	6 miRs + VUC 96/125b/126/145/183/221 0-3/4-7 pos. Markers	4 miRs + VUC 125b/145/183/221 0-2/3-5 pos. Markers
SNS	0.731	0.731	0.808	0.846
SPC	0.935	0.957	0.935	0.957
PPV	0.962	0.974	0.966	0.978
NPV	0.606	0.611	0.683	0.733
pLR	11.205	16.808	12.385	19.462
nLR	0.288	0.281	0.206	0.161
ACC	0.793	0.800	0.847	0.880

ACC, accuracy; nLR, negative likelihood ratio; NPV, negative predictive value; pos., positive; pLR, positive likelihood ratio; PPV, positive predictive value; SNS, sensitivity; SPC, specificity.

**Table 6 ijms-21-03814-t006:** The TaqMan microRNA assays used (Thermo Fisher Scientific, Darmstadt, Germany).

miRNA	Assay Name	Assay ID
miR-21-5p	hsa-miR-21	000397
miR-96-5p	mmu-miR-96 (for hsa-miR-96-5p)	000186
miR-125b-5p	hsa-miR-125b	000449
miR-126-3p	hsa-miR-126	002228
miR-145-5p	hsa-miR-145	002278
miR-183-5p	hsa-miR-183	002269
miR-205-5p	hsa-miR-205	000509
miR-210-3p	hsa-miR-210	000512
miR-221-3p	hsa-miR-221	000524
RNU44 (NR_002750) *	RNU44	001094
RNU48 (NR_002745) *	RNU48	001006

* TaqMan microRNA control assays.

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
