# Peer review of "Urinary MicroRNAs as Potential Markers for Non-Invasive Diagnosis of Bladder Cancer"

_ijms, 2020, doi:10.3390/ijms21113814_

Round 1

Reviewer 1 Report

This manuscript describes urinary miRNA profiling in relation to the diagnosis of bladder cancer. The study appears to have been performed with appropriate scientific rigour and the authors have applied multiple statistical analyses to results, proposing a diagnostic potential of miRNA analysis, in particular in conjunction with cytopathology results (voided urine cytology). Urinary miRNA analysis is not a novel approach to bladder cancer diagnosis or monitoring and a number of prior studies have found correlations between individual or subsets of miRNAs and the presence of primary or recurrent bladder tumours. While interesting, discrepancies between study findings has raised doubts about the robustness or reproducibility of methods, in particular when either elevated or reduced levels of individual miRNAs, and significantly or non-significantly altered levels of the same miRNA(s) are documented in different studies. The investigators have (appropriately) described some of these studies, including where their own results differ from previous reports, however the frequency of divergent findings between studies does reduce potential clinical applications of the findings and of the methods in general.

  1. Based on commentary in the Results section of the manuscript, it is evident that bladder cancer patients had multiple tumours in some cases (documented for cases where Tis co-occurred with pTa, pT1 or pT2, but not for multiple tumours of other stages). In what proportion of cases did this situation arise and were miRNA levels related to total tumour burden (tumour volume) rather than just tumour stage or grade?
  2. What cells do the authors propose to be the origin(s) of the urinary miRNAs detected in their study? If individual miRNAs are being released by tumour cells, this would be consistent with miRNA levels that increase as tumour stage or tumour burden increases. However, what explanation would the authors propose for miRNAs that progressively decrease in comparison to controls?
  3. I am not sure that the manuscript conclusions accurately depict the potential future clinical use of miRNA screening as pathology remains pivotal for definitive bladder cancer diagnosis and staging. Furthermore, it is highly unlikely that clinical investigations for a putative bladder cancer diagnosis would cease following negative results from miRNA screening. Even in the case of miRNA screening results that indicated bladder cancer, the patients would be followed up with cystoscopy in order to obtain cells for definitive diagnosis and staging.

The following are minor comments or typographical, grammatical and syntax errors.

  1. Page 1, lines 35-37: The authors are using 2012 bladder cancer statistics. More up-to-date statistics should be used.
  2. Page 1, line 39: “but is the 4th most common newly diagnosed cancer in Europe and the United States [2,3]. Initial BCa diagnostic steps include voided urine cytology (VUC) as the current standard for non-invasive BCa detection and cystoscopy, an invasive procedure.
  3. Page 1, line 41: “during transurethral resection…”
  4. Page 2, line 48: “is very low at 12-26%”
  5. Page 2, line 49: “test for non-invasive BCa”   “recommended for routine use in clinical practice”
  6. Page 2, line 53: “which eventually allow the number of invasive, inconvenient and expensive cystoscopies to reduce, is ongoing”
  7. Page 2, line 55: “Numerous studies have been aimed at the identification of mRNA expression signatures in tumor tissues and tumor-derived urine specimens that reflect the presence and aggressiveness of BCa and additionally characterize the most powerful predictive biomarker combinations”
  8. Page 2, line 60: “that function as…”
  9. Page 2, line 62: “miRNAs also play”
  10. Page 2, line 62: “such as tumor onset and progression, “
  11. Page 2, line 72: “meta-analyses, nine miRNAs fulfilling the abovementioned criteria”
  12. Page 2, line 78: “as an adjunct to VUC”
  13. Page 3, lines 95 and 98: The word “included” should be deleted.
  14. Page 8, line 212: “to timely start a suitable therapy” should be “to start a suitable therapy in a timely manner” or “to commence treatment in a timely manner”
  15. Page 8, lines 214-216: Nevertheless, none of these new biomarkers is thus far recommended for routine use in the early detection or follow-up of BCa or for replacement or reduction of the invasive procedure of cystoscopy [4].
  16. Page 8, line 217: “reliably predict”
  17. Page 8, line 227: “was finally achieved by use of four of these miRNAs”
  18. Page 8, line 236: “prior to TUR-B”
  19. Page 8, line 237: “prior to tumor resection”
  20. Page 9, line 259: “was also described in several other studies [43-45]. miRNA-183…..”
  21. Page 9, line 260: “a possible marker”
  22. Page 9, line 264: “as a urinary marker has been reported”
  23. Page 9, line 265: Delete “Already”
  24. Page 9, line 266: Delete “the” (last word in line)
  25. Page 10, line 295-296: “profoundly influence the results obtained”
  26. Page 10, line 300: “could reach an increase” should be “reached an increase”
  27. Page 10, line 311: “whereof” should be “although”
  28. Page 10, line 317: “as a further alternative”
  29. Page 11, line 356: “which served as the reference standard”
  30. Page 11, line 362: “As a reference method”
  31. Page 11, line 383: “was employed”
  32. Page 11, line 384: “by the use of “ should be “using”
  33. Page 11, lines389-390: “(each at a final concentration of 0.6-fold).” Also note that “0.6-fold” is not a concentration. This should be corrected to indicate the actual concentration.
  34. Page 12, line 396-397: “Each qPCR with a final volume of” (PCR stands for polymerase chain reaction, so there is no need to repeat the word ‘reaction’)
  35. Page 12, line 398: delete “the”
  36. Page 13, line 441: “Prospective studies are required in order to reveal”
  37. Page 13, line 442: “as a potential tool”

Reviewer 2 Report

In the present study the authors investigate nine urinary microRNAs (derived from the literature) as potential markers for non-invasive diagnosis of bladder cancer. The study is therefore o biomarker validation study. Overall, the study is clearly written and results are of certain interest. There are however some aspects deserving further clarification.

Major:

1) The authors investigate nine different miRNAs making comparisons between patients and controls and according to tumor grade and stage. Bonferroni's correction for multiple comparisons should be performed and only data surviving the multiple comparison corrections should be considered statistically significant.

2) There is no mention about patient's therapy at sampling. Several drugs are able to modify the epigenome. Authors should describe all the therapies performed by the patients (chemo-, radio-, immune- etc.) and see if they are related or not to the expression levels of the selected miRNAs.

3) Commercially available tests based on DNA methylation signatures are already available for bladder cancer (AssureMDx, Bladder CARE, Bladder EpiCheck). The authors should make some comments about them and compare their performances with those of the present miRNA panel

Minor:

The manuscript would benefit from English editing

Reviewer 3 Report

Given that the premise of the study was that VUC does not have high sensitivity for the detection of low-grade BCa, the study does not seem to have achieved what they set out to do. Nevertheless, the study is scientifically sound and well-written. The statistical analyses are appropriate for the study. A few caveats to the conclusions: 

1) It is not clear whether the authors controlled for the potential sources of the urinary microRNAs. 

2) Given that urine biomarker levels may differ with the time at which the urine is collected, it is not clear if this aspect was taken into consideration. 

3) Co-morbidities may also alter the levels of urinary miRNAs. Was this taken into consideration?

4) Some effort should be made to assess the utility of miRNA combinations in detecting low-grade BCa. 

Round 2

Reviewer 2 Report

The authors have nicely addressed my previous concerns